

# Pre-collapse motion of the February 2021 Chamoli rock-ice avalanche, Indian Himalaya

Maximillian Van Wyk de Vries[1,2], Shashank Bhushan[3], Mylène Jacquemart[4,5],
César Deschamps-Berger[6], Etienne Berthier[7], Simon Gascoin[6], David E. Shean[3], Dan H. Shugar[8], and
Andreas Kääb[9]

[1]St Anthonys Falls laboratory, University of Minnesota, Minneapolis, MN, USA
[2]Department of Earth and Environmental Sciences, University of Minnesota, Minneapolis, MN, USA
[3]Department of Civil and Environmental Engineering, University of Washington, Seattle, WA, USA
[4]Laboratory for Hydraulics, Hydrology, and Glaciology (VAW), ETH Zurich, Zurich, Switzerland
[5]Swiss Federal Institute for Forest, Snow, and Landscape Research (WSL), Birmensdorf, Switzerland
[6]CESBIO, Université de Toulouse, CNRS, CNES, IRD, INRAE, UPS, Toulouse, France
[7]LEGOS, Université de Toulouse, CNES, CNRS, IRD, UPS, Toulouse, France
[8]Water, Sediment, Hazards, and Earth-surface Dynamics (waterSHED) Lab, Department of Geoscience, University of
Calgary, Canada
[9]Department of Geosciences, University of Oslo, Oslo, Norway

**Correspondence:** Maximillian Van Wyk de Vries (vanwy048@umn.edu)

**Abstract.** On the 7th of February 2021, a large rock-ice avalanche triggered a debris flow in Chamoli district, Uttarakhand, India, leaving over 200 dead or missing. The rock-ice avalanche originated from a steep, glacierized north-facing slope. In this work, we assess the precursory signs exhibited by this slope prior to the catastrophic collapse. We evaluate monthly slope motion from 2015 to 2021 through feature tracking of high-resolution optical satellite imagery. We then combine these data

5    with a time series of pre- and post-event DEMs, which we use to evaluate elevation change over the same area. Both datasets show that the 26.9 Mm$^3$ collapse block moved over 10 m horizontally and vertically in the five years preceding collapse, with particularly rapid motion occurring in the summers of 2017 and 2018. We propose that the collapse results from a combination of snow-loading in a deep headwall crack and permafrost degradation in the heavily jointed bedrock. Our observation of a clear precursory signal highlights the potential of satellite imagery for monitoring the stability of high-risk slopes. We find that the

10   timing of the Chamoli rock-ice avalanche could likely not have been forecast from satellite data alone.

## 1 Introduction

### 1.1 Landslide hazard

Landslides represent a major geohazard, and cause thousands of deaths each year (Petley, 2012; Froude and Petley, 2018). Preventing or mitigating landslide hazard is a major challenge facing geoscientists and hazard managers. Evaluating landslide

15   hazard is challenging due to the wide range of source conditions and the varying temporal scales at which the driving processes interact. Landslides are also associated with a wide range of short- to long-term triggers, ranging from earthquakes to water





flow, or simple weaknesses in the rock, which further complicates their forecasting and process understanding (van Westen et al., 2006).

Ground-based observations of displacement (e.g. with GNSS/GPS, ground-based radar interferometry), tilt (e.g. inclinome-
ters), pressure (e.g. piezometers), and other parameters can be useful in monitoring landslide progression (e.g. Uhlemann et al., 2016). When observed, landslide precursory signs may be used to forecast a failure time or improve monitoring (Federico et al., 2012; Fukuzono, 1985; Intrieri et al., 2019; Wegmann et al., 2003). In many cases the nature and magnitude of these precursory signs precludes their detection in the absence of sensitive equipment. In-situ observations can be sensitive to even small changes in slope properties, and are therefore valuable for the forecasting of instability (Sättele et al., 2015; Stähli et al.,
2015). However, ground-based observations have important limitations: (i) Prior knowledge of a potential slope instability is required in order for the correct instrumentation to be installed in the right locations, (ii) the landslide source regions may be located in inaccessible terrain, preventing the installation of in-situ monitoring equipment, (iii) monitoring systems can be prohibitively expensive and require highly specialized expertise for data evaluation, and (iv) the area that can be monitored is generally limited to individual hillslopes. Altogether, ground-based monitoring techniques are useful for landslide monitoring
in many cases, but are insufficient for monitoring large regions or where a-priori knowledge is lacking.

An increase in satellite data availability and resolution has promoted remote sensing as an alternative or complementary landslide detection and monitoring tool (e.g. Kirschbaum et al., 2019; Dille et al., 2021). Satellite remote sensing may lack the precision of some ground-based monitoring techniques, but but it can provide a low-cost (for the end user) and easily accessible way to monitor vast and inaccessible terrain at daily to weekly temporal and 0.3 to 30 m spatial resolution. Qualitative visual
analysis of satellite imagery allows for the rapid identification of surface changes that may be associated with slope instabilities or the initiation of landslide motion. Further quantitative processing of satellite imagery enables the monitoring of horizontal and vertical land motions – for example via feature tracking or stereographic digital elevation model (DEM) generation. In-terferometric synthetic aperture radar (InSAR) can provide mm to cm-resolution line of sight displacements (e.g. Handwerger et al., 2019; Jacquemart and Tiampo, 2021; Manconi et al., 2018). Growing archives of high-resolution, open access Earth
observation data remain largely untapped for landslide monitoring. In this study we use the data-rich 7 February 2021 Chamoli rock-ice avalanche as a case study for the remote identification of landslide precursory signs. We first introduce landslide haz-ards in the Himalaya with specific focus on the Chamoli event, and then offer a general overview of remote sensing of slope instabilities. Next, we explain the methods used in the current study, and present and discuss the results.

## 1.2 Landslide hazard and risk in the Himalaya

Landslides occur in high mountain areas all over the world. Landslide risk is greatest where zones of high topographic relief intersect with high population densities or infrastructure – which is the case across much of the Himalayan region. Over 50 million people live directly within the Himalaya, with a further 700 million living within associated watersheds (Dimri et al., 2019). A combination of extreme topographic relief, regular tectonic activity, high seasonal rainfall intensities, glacierization, and steep slopes make the Himalaya particularly susceptible to landslides (Kirschbaum et al., 2019).





Several factors have contributed to raising landslide risk across the region in recent decades: first, climatic warming has driven rapid thinning and retreat of Himalayan glaciers – which are currently losing over 10 Gt of mass per year (e.g. Kääb et al., 2012; Brun et al., 2017; Shean et al., 2020; Jakob et al., 2021; Hugonnet et al., 2021). Glacier retreat may contribute to a range of factors conducive to landslides, including a reduction in slope buttressing and an increase in meltwater availability (Holm et al., 2004; Fischer et al., 2006; Huggel et al., 2012; Kos et al., 2016; Coe et al., 2018; Dai et al., 2020a; Glueer et al., 2020). In addition to glacier retreat, permafrost degradation has also been documented to reduce slope stability (Gruber and Haeberli, 2007; Allen et al., 2011; Fischer et al., 2012; Krautblatter et al., 2013; Haeberli et al., 2017; Magnin et al., 2019; Pörtner et al., 2019; Patton et al., 2019; Deline et al., 2021). Second, increasing populations, economic growth, and infrastructure development in high-mountain valleys have greatly expanded the potential consequences of landslides. This second point is apparent for the Chamoli disaster, in which the majority of deaths occurred at hydropower plants that were recently built or were under construction (Shugar et al., 2021). Other factors, including changes in precipitation pattern (e.g. Li et al., 2018; Kirschbaum et al., 2020) and land use (Cummins, 2019) may also contribute to evolving landslide hazard potential.

### 1.3 The 2021 Chamoli hazard cascade

During the morning of 7 February 2021, a 26.9 [95% confidence interval 26.5-27.3] Mm$^3$ wedge of rock and ice detached from the north face of Ronti, a 5500 m peak in the Uttarakhand Himalaya (Fig 1.). This wedge then dropped around 1800 m to the Ronti Gad valley floor, where it continued down-valley towards the Rishiganga and Dhauliganga rivers and transformed into a debris flow (Shugar et al., 2021; Cook et al., 2021). The collapse block was composed of approximatively 80% bedrock and 20% glacier ice. Frictional heat generation calculations suggest that most or almost all of the glacier ice melted during the 3400 m drop from the collapse source to the hydropower stations (Shugar et al., 2021). This melting of the ice faction, combined with major sediment deposition at the confluence of the Ronti Gad and Rishiganga, increased the initial rock-ice avalanche's water content and converted it into a debris flow. The resulting debris flow caused further downstream damage, leaving 204 missing or killed and destroying two hydropower stations.

### 1.4 Remote-sensing techniques

#### 1.4.1 Feature tracking

Optical feature tracking is a versatile technique, which can be used to track surface motion by evaluating the relative position of features or patterns in repeat satellite images or aerial photos. Feature tracking has been applied to a variety of problems, including tracking post-seismic ground deformation (e.g. Leprince et al., 2007), measuring glacier flow velocities (e.g. Bindschadler and Scambos, 1991; Heid and Kääb, 2012; Millan et al., 2019; Van Wyk de Vries and Wickert, 2021), and measuring landslide displacements (e.g. Behling et al., 2014; Lucieer et al., 2014; Manconi et al., 2018; Dai et al., 2020a; Dille et al., 2021).


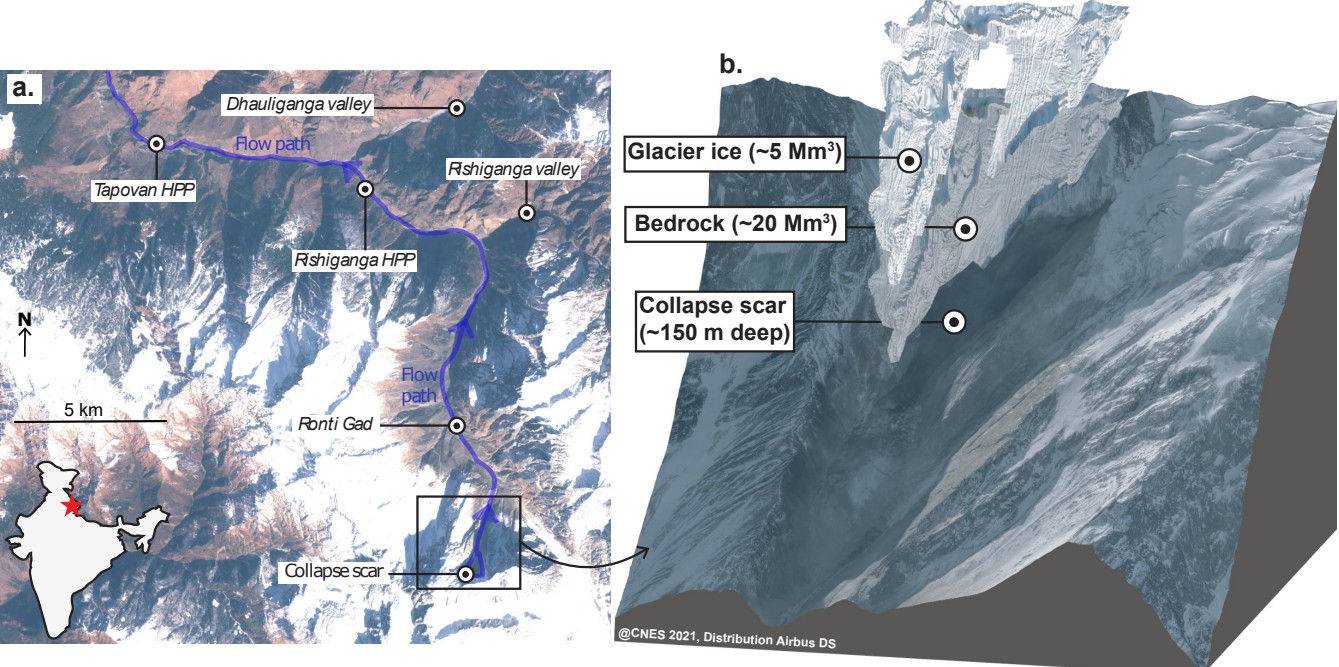

**Figure 1.** The 7th February Chamoli rock-ice avalanche: (a) shows the path of the collapse, along with key locations (HPP refers to hydropower plant) annotated on a 10th of February 2021 Sentinel-2 image (b) 3D visualization shows the post-collapse scar with reconstruction of the overlying bedrock and glacier ice.

### 1.4.2 Stereo-DEM generation

Stereo-DEM generation uses two or more overlapping optical images to reconstruct surface topography. These images are acquired at the same time but from different viewing angles. Software implementing photogrammetric principles can then be used to derive elevation products (such as DEMs) from these images. With the recent availability of very high resolution satellite stereo imagery, these approaches can now be used to generate detailed DEM products over large spatial areas (e.g. Korona et al., 2009; Morin et al., 2016; Shean et al., 2016; Porter et al., 2018; Howat et al., 2019). Repeat DEMs obtained at different time periods can provide precise estimates of surface elevation change associated with many processes, including glacier change (e.g. Brun et al., 2017; Willis et al., 2018; Zheng et al., 2019; Shean et al., 2020), snow accumulation/melt (e.g. Deschamps-Berger et al., 2020; McGrath et al., 2019; Bhushan et al., 2021), volcanic deformation (e.g. Bisson et al., 2021; Schaefer et al., 2012), and landslide or debris flow events (e.g. van Westen and Lulie Getahun, 2003).

### 1.4.3 InSAR

Satellite-based InSAR is a powerful tool for detecting small changes at the Earth's surface from space. It has been widely used to quantify ground displacements caused by processes such as earthquakes (e.g. Massonnet et al., 1993; Barba-Sevilla et al.,


2018), groundwater extraction (e.g. Samsonov and d'Oreye, 2017; Motagh et al., 2017), volcanic unrest (e.g. Rosen et al., 1996; Tiampo et al., 2017), or landslides (e.g. Manconi et al., 2018; Handwerger et al., 2019; Dai et al., 2020b; Mondini et al.,

2021; Jacquemart and Tiampo, 2021). By measuring the shift of the radar phase relative to earlier measurements of the same features, InSAR can provide measurements of ground deformation at millimeter and centimeter scales. Active radar sensors can image the Earth's surface through clouds and darkness, a major advantage over passive optical sensors (e.g. Massonnet and Feigl, 1998). Leveraging InSAR data for the detection and assessment of mass movements, however, is not without challenges. The oblique viewing geometry of radar satellites means that radar data can be rendered useless in areas of steep topography due

to the effects of shadowing, foreshortening, and layover (Massonnet and Feigl, 1998; Wasowski and Bovenga, 2014). Finally, in case of rapid displacements that surpass the phase-aliasing thresholds or dramatic changes in the surface cover or geometry, a loss of interferometric coherence can prohibit the quantification of (the full) ground deformation (Manconi, 2021). Despite these drawbacks, many studies have shown that InSAR can be successfully applied to assess stability of slopes even in high relief terrain (e.g. Manconi et al., 2018; Handwerger et al., 2019; Bekaert et al., 2020; Jacquemart and Tiampo, 2021).

**1.5 Objectives**

The objective of this study is to evaluate the pre-collapse conditions of the 7 February 2021 Chamoli rock-ice avalanche, in particular:

1. What was the scale and geometry of pre-collapse surface change, and what insight do these changes provide into the collapse mechanisms?

2. Would these pre-collapse datasets and tools would be adequate to identify this hazardous slope without the prior knowledge of its failure?

**2 Methods**

We used a range of datasets and processing workflows to investigate the pre-collapse conditions of the Chamoli rock-ice avalanche:

1. Optical satellite imagery (Landsat and Sentinel-2) was used to investigate visible changes in the collapse region over the years to decades prior to the rock-ice avalanche

2. Feature tracking of optical satellite imagery (Sentinel-2, Planet, Cartosat-1, and SPOT7) was used to derive horizontal displacements

3. Digital elevation models (DEMs) from optical satellite stereo-imagery (WorldView-1/2/3,GeoEye-1, Pleiades-HR, SPOT-

7 and Cartosat-1) were used to derive vertical changes

4. Sentinel-1 C-band radar imagery was used to calculate interferometric synthetic aperture radar (InSAR) displacement maps



## 2.1 Qualitative observations of slope change

We investigated three decades of pre-collapse optical satellite imagery to gain a preliminary understanding of pre-landslide
changes. We documented changes in the north-facing slope of Ronti peak, which sourced the February 2021 rock-ice avalanche, using all available data from Landsat 5 (TM), Landsat 7 (ETM+), Landsat 8 (OLI), and Sentinel-2 with a cloud cover of less than 60%. We focused our observations on surface changes, including deformation and fracturing, and rock or ice avalanches originating from the collapsed block or surrounding area.

   Our ability to detect change is limited by the spatial resolution of the imagery used (15-30 m for Landsat and 10 m for
Sentinel-2). We examined a 31-year (1990-2021) time series of satellite imagery (Fig 2), including 122 Landsat 5 images, 43 Landsat 7 images, 34 Landsat 8 images, and 155 Sentinel-2 images. A full list of images is provided in the supplementary material, along with a brief description of any anomalous features.

### 2.1.1 Optical feature tracking

We used feature tracking with a range of medium (10 m) to high (2.5 m) resolution satellite imagery to evaluate the pre-collapse
motion of the Ronti peak north slope. We used two different feature-tracking toolboxes: GIV (Van Wyk de Vries and Wickert, 2021) and AutoRIFT (Lei et al., 2021). Both GIV and AutoRIFT are based on three core components: a pre-processing module which applies one or more filters to images to enhance distinct surface features for tracking, a multipass 2D image correlator, and a post-processing module to identify and filter erroneous displacement values (Van Wyk de Vries and Wickert, 2021; Lei et al., 2021). The GIV toolbox is written in MATLAB and performs image cross correlation in the frequency domain, while
AutoRIFT is written in python/C++ and performs the cross correlation in the spatial domain. Using GIV, we pre-processed the imagery using an orientation filter and ran the cross-correlation with a reducing window size from 20 to 5 pixels and a window overlap of 50%. In AutoRIFT we pre-processed the imagery with a Laplacian filter and used adaptive window sizes between 32 and 64 pixels with a skip rate of 8 pixels for the cross-correlation.

   We calculated velocities using all available Sentinel-2 images through February 2021, excluding any images with a local
cloud cover greater than 60% (based on the L1-C QA band cloud mask). A total of 155 images were available, for a total of 5237 image pairs with a time separation between 50 and 500 days. We processed these image pairs using GIV. We also resampled the velocity timeseries to monthly resolution (see Fig 4 c-g) using a weighted averaging scheme described in Van Wyk de Vries and Wickert (2021).

   We downloaded all PlanetScope Dove Classic (4-band) Level-1B imagery with less than 20% cloud cover acquired between
January 2020 and January 2021. We processed 4701 image pairs using AutoRIFT with time separation of 100 to 350 days. The Near-infrared (NIR) band from the L1B images was orthorectified on the 2015 pre-event reference DEM (Bhushan and Shean, 2021) and the systematic median offset (computed over static, non-glacierized surfaces) was removed from each pairwise surface displacement map in both E-W and N-S directions. Despite the higher product resolution (3 m vs 10 m Sentinel-2 images) and use of a high-resolution DEM for improved orthorectification, the Planet velocity maps had a high random
background noise. We attribute this to spurious correlation over surfaces with varying shadow cover due to steep slopes and



changing illumination, as the images were captured by different satellites during different times of the day/year. To compensate for this higher background noise, we chose a higher minimum temporal separation between Planet image pairs when calculating time-averaged velocity maps. We also calculated displacements (using both GIV and AutoRIFT) on one pair of high-resolution Cartosat-1 images (Oct 2017 to Nov 2018).

We used this velocity data to evaluate whether the collapsed block moved prior to collapse – with a null hypothesis that the block moved no more than the surrounding 'stable' (non-glacierized) bedrock. A medial bedrock ridge near the center of the collapsed block provides motion of the underlying rock, rather than simply flow of the overlying glaciers. We divided the collapse block into three different regions alongside a zone of stable ground, and create a time series of average displacement for each zone.

## 2.2   DEM generation

We produced multiple pre-event and post-event DEM products from very high-resolution (Maxar/DigitalGlobe WorldView-1/2/3, GeoEye-1 and Airbus/CNES Pleiades, 0.3 to 0.5 m GSD) and high-resolution (Airbus SPOT-7 and ISRO CartoSat-1, 1.5 m to 2.5 m GSD) satellite imagery captured between 2015 and February 2021. The DEM products were used to calculate the vertical motion of the collapse block from 2015 to February 10, 2021.

We used the NASA Ames Stereo Pipeline (Shean et al., 2016; Beyer et al., 2018) to process all of the images. For this particular study, we primarily used four products spanning two time periods: the 2015 pre-event WorldView DEM composite (Bhushan and Shean, 2021); an intermediate period, a 2018 pre-event DEM composite produced by averaging the November 2018 CartoSat-1 (Appendix A1) and December 2018 SPOT-7 (Appendix A2); and the February 10-11, 2021 post-event composite DEM derived from Pleiades and WorldView/GeoEye stereo imagery (Shean et al., 2021). We calculated the difference between three composite DEM products to create 2015-2018, 2015-2021, and 2018-2021 DEM of difference (DoD). The first DoD provides insight into vertical changes in the hillslope prior to failure, while the second DoD provides the volume and geometry of the collapsed block. We calculated an empirical uncertainty estimate for each DoD using the tiling method (Berthier et al., 2016; Miles et al., 2018; Jacquemart et al., 2020).

## 2.3   InSAR maps

We analyzed Sentinel-1 data from the ascending and descending orbit tracks 56 and 63, respectively, to investigate whether the precursory motion of the collapse block could have been detected from radar interferometry. All radar data was downloaded from the Alaska Satellite Facility Distributed Active Archive Center (ASF DAAC). Because the descending track is heavily affected by layover artefacts, we only performed the full processing with data from the ascending orbit. We processed the data with the InSAR Scientific Computing Environment (ISCE; Rosen et al., 2012), removed the topographic phase using

the 2015 pre-event WorldView DEM (Bhushan and Shean, 2021) composite (resampled to 8m), and masked out all pixels with an interferometric coherence of less than 0.3. Single look complex (SLC) images were multi-looked to 1 and 3 looks in azimuth and range, respectively. We generated 108 interferograms covering the period of January 2017 to November 2020,





each spanning 12 days. We manually selected the best interferograms and performed unwrapping with the Statistical-Cost, Network-Flow Algorithm for Phase Unwrapping (SNAPHU; Chen and Zebker, 2002).

## 3 Results

Where observation was possible, the different methods are in agreement: the slope fractured and was displaced on the order of tens of metres prior to the eventual collapse. The 2021 rock-ice avalanche was also preceded by several other large avalanches – although these were primarily sourced from an adjacent hanging glacier.

### 3.1 Qualitative observations of slope change

We identified four main types of processes in our 31 year optical satellite image time series:

1. Major ice avalanches (01-04/2000 and 09-10/2016): Large-volume ice avalanches, which originated from the steep hanging glacier to the west of the collapse block. These temporarily filled Ronti Gad with ice, snow, debris and sediment.

2. Minor snow or ice avalanches (2005, 2006, 2007, 2008, 2012, and 2015): Smaller volume avalanches, which may either have originated from the adjacent hanging glacier or the seasonal snowpack. These did not appear to infill the underlying valley with any significant quantity of material (with the exception of one ~500 m long snow/ice deposit in May 2006).

3. Minor landslides avalanches (2007, 2009, 2011, 2012, 2013, and 2015): Minor rockfalls or rock avalanches originating from Ronti peak, or the weak sediment on the flanks of Ronti Gad. These also do not appear to have deposited major volumes of sediment.

4. Opening and widening of cracks at the headwall of the collapse block (2016-2021): Gradual opening of a wide crack in the north face of Ronti peak.

We only interpret the 4th process type (crack opening) is a real sign of pre-collapse conditions. Minor rockfalls and snow/ice avalanches are a common feature of high relief, high slope active landscapes. The major ice avalanches represent a serious geohazard in the upper Ronti Gad, but appear to relate to internal dynamics of the western hanging glacier rather than instability in the underlying bedrock. The area of these 2000 and 2016 major ice avalanches was estimated at 0.16 km$^2$ and 0.2 km$^2$, with melting and/or redistribution of the the resulting valley floor deposits within three years of the event (Shugar et al. (2021); Supplementary section 3.1). Regular large ice avalanches have been observed at many other hanging glaciers in active, high-mountain environments (e.g. Faillettaz et al., 2008; Vincent et al., 2015).

The conspicuous crack at the headwall of the failed block was first visible in optical imagery in March 2016, although its location roughly aligns with a pre-existing glacier crevasse – suggesting that minor crack opening in the bedrock may have preceded this date. The crack grew to approximately its maximum size by the end of 2018, and appeared to reduce in depth or become infilled over the course of 2019 and 2020. The crack widened further between 2018 and the 7 February 2021 collapse,





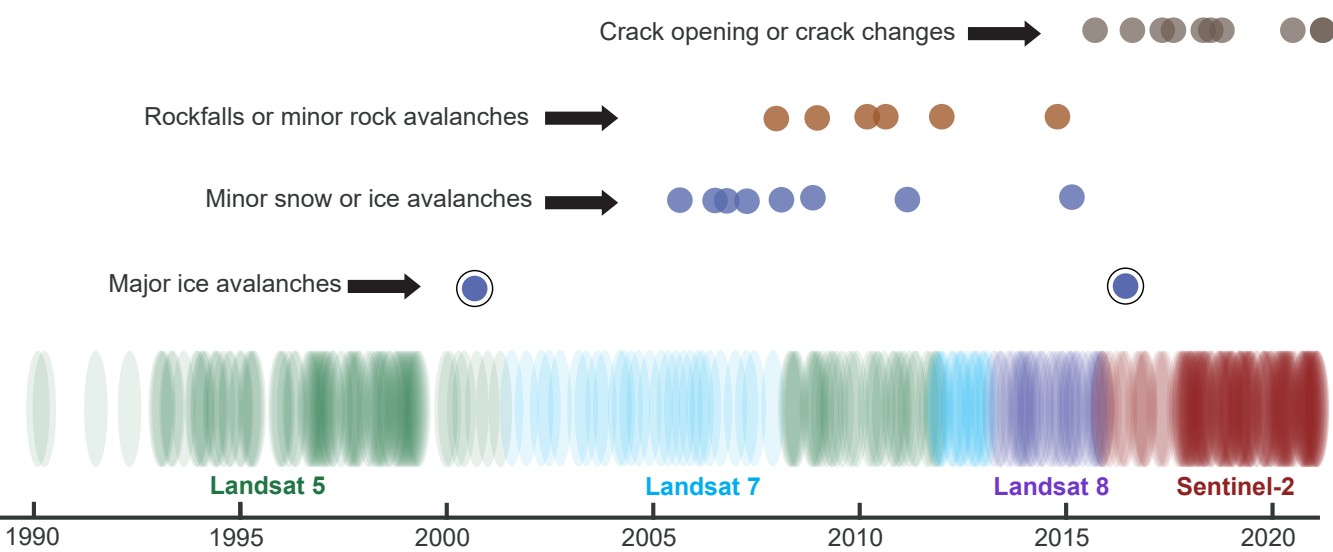

**Figure 2.** Timeline of images analysed for change at the Chamoli site prior to the 7th of Ferbuary 2021 collapse, with major events or changes seen over this period. The first image (06/02/1990) was taken 31 years before the collapse, and the last image (05/02/2021) two days before the collapse.

but less rapidly than the opening in 2016-2018. We confirmed these observations with several very-high ( 0.5 m) resolution images (Fig 3).

### 3.2 Optical feature tracking

Feature tracking provides the most complete spatio-temporal assessment of displacement of the methods used in this study – with data coverage from late 2015 until early 2021. We used results from the Cartosat-1 image pair and the Planet archive for validation of the Sentinel-2 displacements. In all three cases, the collapsed block (most notably, the bedrock ridge at the centre of this block) exhibited displacements exceeding the background noise level on stable bedrock ($<1\,\mathrm{m\,yr^{-1}}$).

The horizontal velocity of the collapsed block ranged from around $5\,\mathrm{m\,yr^{-1}}$ to $20\,\mathrm{m\,yr^{-1}}$, with the most rapid motion 225 occurring in the summers of 2017 and 2018 (see Fig 4 d-g). We do not observe an increase in velocity of the collapsed block immediately prior to its failure in February 2021. The Sentinel-2 image record includes 7 cloud-free images from early 2021, including one image taken two days prior to the collapse, therefore this lack of speed-up is unlikely the result of a temporal data gap. Periods with the highest block velocity correspond to periods of greatest increase in headwall crack width – particularly the summers of 2017 and 2018. This is consistent with motion occurring on the entire collapsed block, rather than only on the 230 glaciers or a superficial layer of rock.

Total 2016-2021 horizontal displacements were  20-30 m (Fig 4a), of similar magnitude to the width of the crack as measured directly from Sentinel-2 imagery. Projecting these horizontal displacements onto the steep surface slope (mean of 42.6°) results



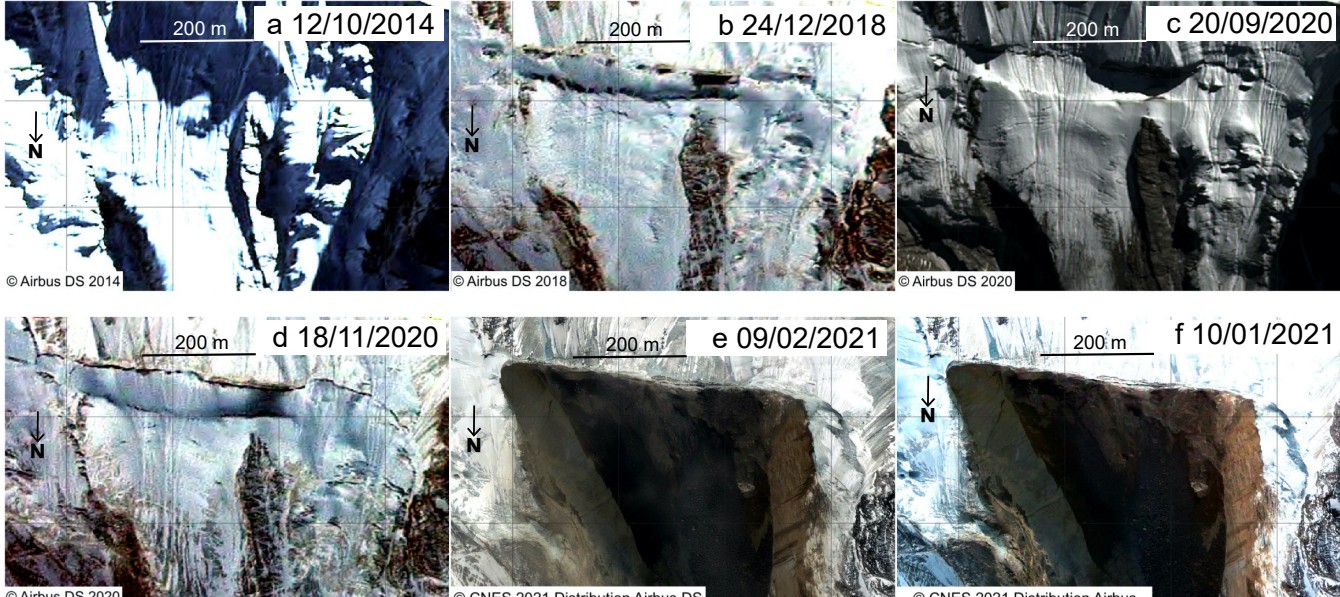

**Figure 3.** Time series of headwall crack opening in high-resolution optical images from SPOT-7 and Pleiades-HR.

in an apparent increase of $\sim$ 36 %, or $\sim$ 25-40 m. Overall, the feature tracking results demonstrate that the collapse block was mobile several years prior to its collapse in 2021.

## 235   3.3   DEM analysis

We calculated the geometry of the collapsed block, equal to the zone of negative elevation change in the 2018-2021 DoD (Fig 5c; volume = 26.9 [95% confidence interval 26.5-27.3] Mm$^3$ ; Shugar et al. (2021)). The earlier DoD (2015-2018) shows a very different pattern (Fig 5b), with a  100 m wide zone of elevation loss at the upper altitude limit of the collapsed block ('headwall crack') and a broad zone of elevation gain over the remainder of the block ('bulge'). The magnitude of this pre-collapse

elevation loss is greatest in the central and western portion of the headwall crack, while the elevation gain is most pronounced on the central and eastern portions of the bulge. The DoD uncertainties scale inversely with the size of area (number of pixels) considered: $\pm$4.2 m, 1.7 m, 7.3 m, and 2.7 m (10 by 10 m); $\pm$3.0 m, 0.8 m, 4.7 m, and 1.8 m (50 by 50 m); $\pm$2.4 m, 0.2 m, 3.8 m, and 1.1 m (250 by 250 m) for the 2015-2018 Cartosat, 2015-2018 SPOT, 2018-2021 (SPOT) and 2015-2021 DoDs respectively.

DEM analysis further confirms the results from direct image observations and feature tracking – large changes occurred on the collapsed block prior to its collapse. The zone of negative elevation change is wider than the crack as directly observed in optical imagery, which may result from limits in the DEM resolution or partial collapse of the surrounding rock or ice into the crack.

**Figure 4.** Surface displacement and horizontal velocity from optical image feature tracking. (a) shows the total displacement over the entire Sentinel-2 era, (b) shows a snapshot velocity during an episode of rapid displacement in Summer 2018, and (d)-(g) show time series of velocity averaged across specific zones shown in (c). Note the episodes of rapid displacement in 2017-18 relative to 2016 or 2020, corroborated by the Cartosat-1 and Planet derived velocities.



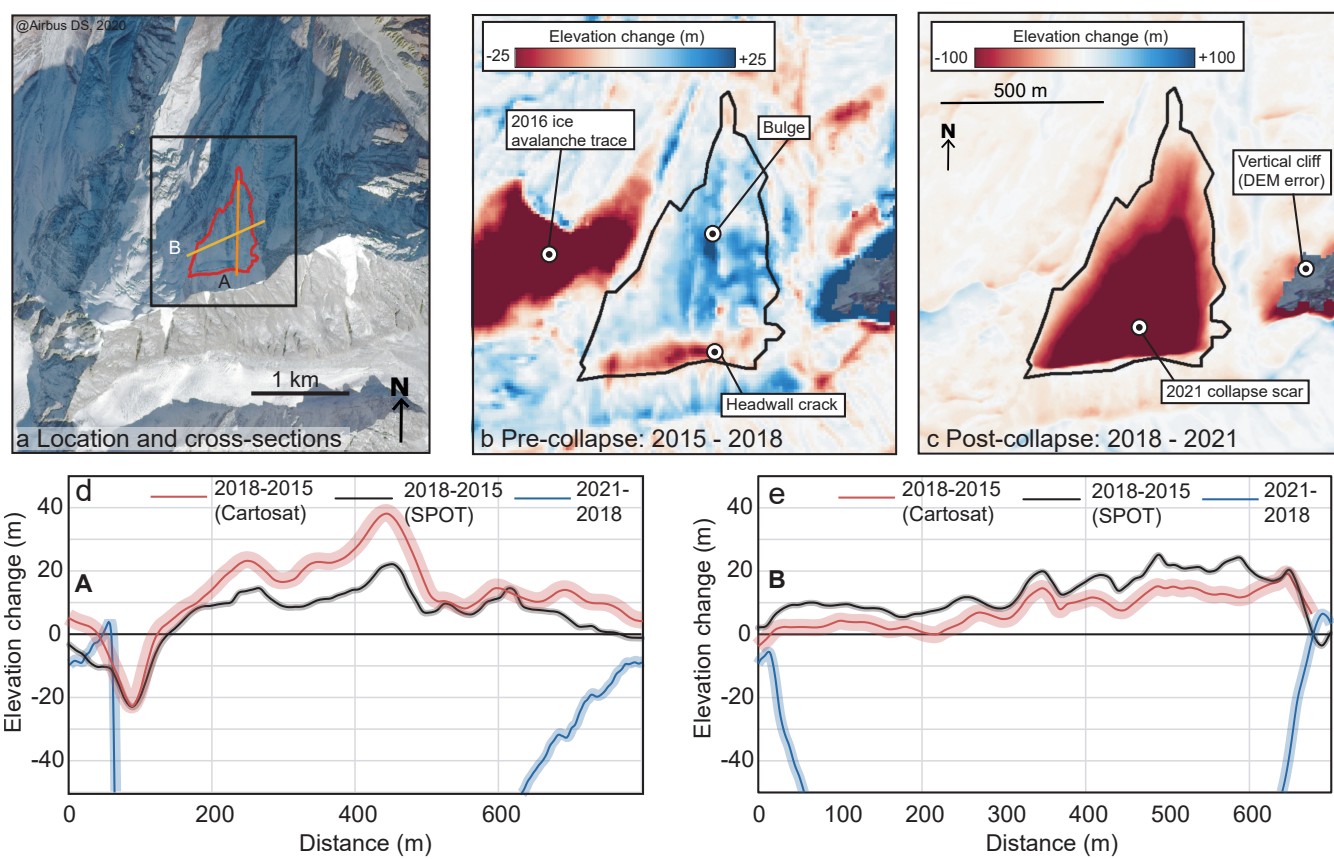

**Figure 5.** Elevation change of the avalanche zone pre- and post-collapse. (a) SPOT-7 true color composite image from September 2020 with location of cross-sections A and B for context, (b) and (c) provide DoD maps of different time periods, while (d) and (e) show two cross-section profiles (a) across the DoD. Cross-sectional uncertainties are assigned for an area equal to the length of the section line multiplied by the pixel size.

## 3.4 InSAR maps

Even with knowledge of the location of the failed block, the processed interferograms do not allow for a pre-collapse identification of the instability on Ronti Peak. Of the 108 available interferograms, roughly half exhibited a complete loss of coherence, largely due to snow cover (November through May). Good quality interferograms are limited to summer months, and on the collapse block, coherence is only retained on the ice free part at the bottom of the wedge. The upper, glacier-covered part of the collapse block remains decorrelated, likely due to shadowing and glacier/snow cover. Figure 6a highlights the very low radar

backscatter in this zone, and Figure 6b/c confirms the spatial agreement between the loss of coherence and glacier cover. Data gaps lower in the valley are also related to loss of coherence, possibly due to vegetation cover or moisture variability. Many interferograms are characterized by high amounts of noise, likely from variable atmospheric properties .





A high quality interferogram from July 2020 (Fig. 6b) does not indicate any motion on the collapse block in the summer prior to the failure, but this cannot be assessed in other interferograms due to high noise levels. In less steep terrain north-west

of the collapse block, the motion of a rock glacier (on the order of $\mathrm{cm\,yr^{-1}}$) can consistently be detected in the interferograms (Fig 6b). Despite its sensitivity, InSAR is not able to provide any conclusive information about the pre-failure conditions of the collapse block in this challenging terrain.

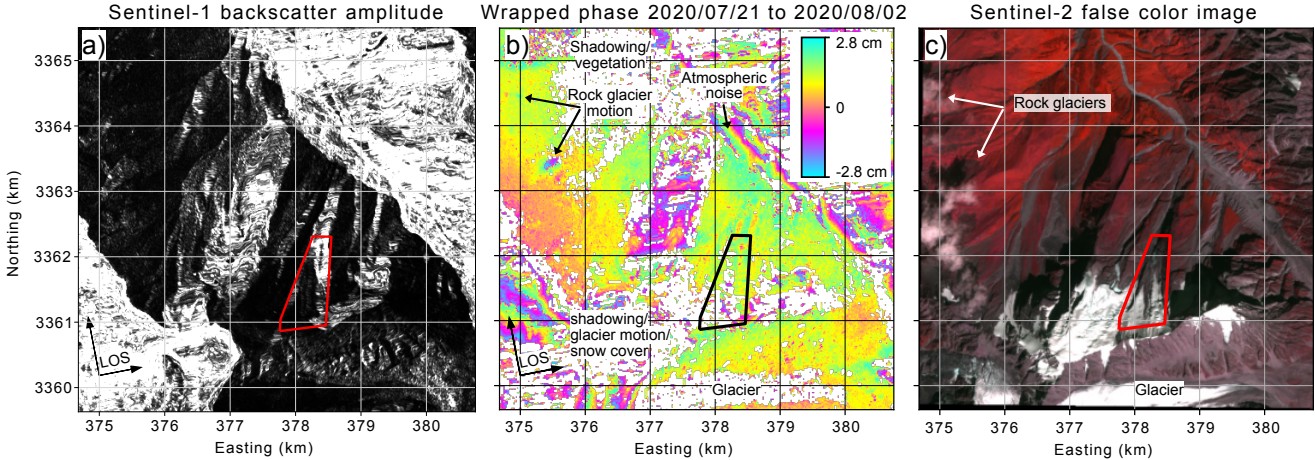

**Figure 6.** Sentinel-1 radar backscatter amplitude from the ascending orbit (a), wrapped phase (0 to $2\pi$) ascending orbit interferogram from July 2020 (b) and corresponding false color image (c). Large areas of low coherence (masked as white) and patchy coverage illustrate the complexities of InSAR monitoring in high alpine terrain. The avalanche block is outlined in black and red.

## 4 Discussion

The pre-collapse motion of the avalanche block raises important questions about the causes and timing of the slope failure. In

this section, we explore the answers to these questions using our multi-dataset observations, and then discuss the potential and limitations of satellite data for remote hazard monitoring.

### 4.1 Three-dimensional block motion

We examined the three-dimensional motion of the collapse block as a first step towards understanding the Chamoli rock-ice avalanche collapse mechanism(s). Rotation and translation are the two primary modes of landslide motion (e.g. Záruba and

Mencl, 2014), with each having a distinct surface displacement pattern. We used a combination of horizontal displacement (feature tracking), vertical displacement (2015-2018 DoD), collapse block thickness (2018-2021 DoD), and post-landslide topography to calculate the dominant mode of pre-collapse motion for the Chamoli collapse block.

We compared our observations of vertical and horizontal slope displacement to a synthetic displacement, with the hypothesis that all of the observed change could be explained by translation. To model this translation, we set the direction of motion to





that of the steepest slope (∼NNE) and its magnitude to 20 m. The displacement magnitude is chosen to match our observed

horizontal displacement from feature tracking - and is consistent with the findings of Qi et al. (2021).

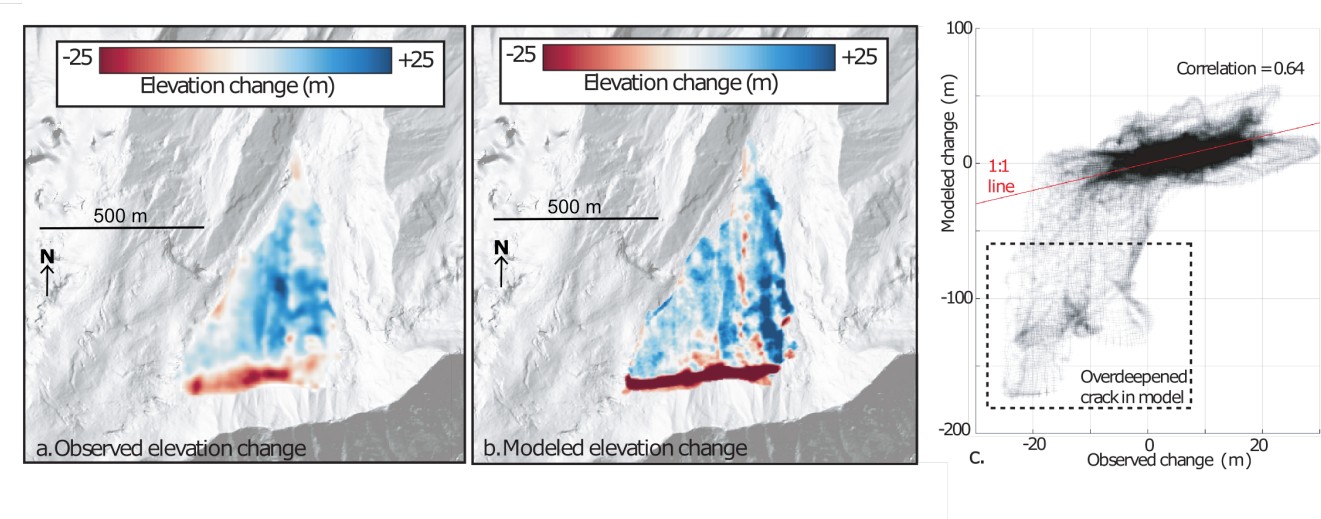

**Figure 7.** (a) Observed and (b) modeled elevation change of the Chamoli landslide block prior to collapse. The modeled scenario (b) is based
on 20 m of pure downslope translation. (c) shows a scatterplot comparing each observed and modelled pixel.

Figure 7 shows a comparison between the observed change in surface elevation of the landslide block, and the modeled
change. The pattern of elevation change is similar for the observed and modeled cases – both exhibit a deep summit crack,
bulging in the lower collapse zone, and greater elevation gain on this bulge to the east relative to the west. The 2D correlation
score is 0.64, with the greatest model-data difference at the headwall crack, which is as much as 150 m deeper in the model
case. These results are consistent with the Chamoli collapsed block moving downslope by translation in the years prior to
collapse.

### 4.2  A possible avalanche triggering mechanism

A viable triggering mechanism for the Chamoli landslide must explain both the lag between the initial instability and collapse,
and the timing of the collapse – in the middle of the winter. Syn-collapse seismic signals show that there was no seismic trigger
for the collapse (Pandey et al., 2021; Shugar et al., 2021; Cook et al., 2021). Nearby meteorological stations and reanalysis
data reveal heavy snowfall and a  5 K positive temperature anomaly in the week preceding collapse, as well as a temperature
inversion in the valley (e.g. Pandey et al., 2021; Dandabathula et al., 2021; Zhou et al., 2021; Shugar et al., 2021). On the
longer term, this region has warmed ∼0.14 K per decade (Qi et al., 2021; Shrestha et al., 2021).
Zhou et al. (2021) and Dandabathula et al. (2021) propose that this sudden temperature increase may have triggered the
collapse, and Rana et al. (2021) associates it with lubrication of pre-existing fractures via melting of fresh snow. Kropáček





et al. (2021) and Pandey et al. (2021) suggest that loading from heavy snowfall may have contributed to the failure. Despite the positive temperature anomaly, temperatures at the collapse altitude ( 5000 m) would have been below freezing point on the day of collapse, and liquid water would not have been present at the surface (Shugar et al., 2021; Dandabathula et al., 2021). Positive summer temperatures (Shrestha et al., 2021) and a steep surface slope of the collapse block will have prevented strong cumulative surface loading of the collapse block through snow deposition. Existing hypotheses do not provide strong mechanistic links between observed meteorological changes and the slope failure.

The stability of a slope can be described by the balance between two terms: driving forces ($F_D$) and resistive forces ($F_R$). Driving forces are primarily gravitational, while resistive forces are primarily related to slope cohesion and friction. For a detached wedge such as the Chamoli collapse block, dominant resistive forces are likely friction along the margins and base of the collapsed block. The balance between these two forces is known as the factor of safety $FS$:

$$FS = \frac{F_R}{F_D} \tag{1}$$

A slope is considered unstable when its factor of safety falls below 1 (e.g. Záruba and Mencl, 2014; Das and Sivakugan, 2016).

The Chamoli collapse area is composed of heavily jointed bedrock (e.g. Shugar et al., 2021). A pure translational pre-collapse motion is consistent with a collapse block basal shear plane along a single bedding plane. High-resolution post-collapse satellite imagery also suggests that the detachment occurred along a bedding plane. This failure plane may have been superficially weakened by freeze-thaw fracturing (Qi et al., 2021; Kropáček et al., 2021; Shrestha et al., 2021), or at greater depth by changes in permafrost conditions (e.g. Gruber and Haeberli, 2007; Krautblatter et al., 2013). The surface velocity peaks in summers 2017 and 2018 suggest that surface meltwater may have reached into the later failure surface. Meltwater infiltration may directly impact friction ($F_D$), and in a delayed way also alter ground temperatures through advection of heat and release of latent heat upon refreezing. Gruber and Haeberli (2007) note that advection-driven melt of permafrost thaw corridors may drive destabilization of large volumes of rock. Deep permafrost thaw may occur over long timescales (e.g. Gruber and Haeberli, 2007; Krautblatter et al., 2013), and provides one potential explanation for the 5-year lag between initial instability and collapse.

The deep headwall crack provides accommodation space for cumulative snow accumulation and loading, and also limits the melting of accumulated snow by reducing its surface exposure. Observations of elevation change over 2015-2018 show the opening of a crack at least 25 m deep at the collapse block headwall (Fig 7a), although DEMs may underestimate the true depth of the crack due to viewing angle, slope geometry, and stereo DEM processing parameters. The purely translation model of block motion (Fig 7b) suggests that the true crack depth would have been closer to 150 m. Snow, ice, or rock debris loading within a headwall crack would exert a horizontal force on the collapse block. This horizontal force ('push') acts to reduce the factor of safety both by directly increasing the driving force of the collapse block, and reducing the angle between the driving force vector and slope direction (equivalent to an increase in slope, see Appendix B).





Accumulation of snow or ice in the crack is visible in optical satellite imagery, with additional input from snow/ice
avalanches from the overlying slope (e.g. Fig 3b-d). A storm in the days preceding the 7th February collapse brought substantial snowfall to the Chamoli region, with local snowfall estimates ranging from 8.5 to 48 mm water equivalent of precipitation (Shugar et al. (2021); estimates from local weather stations and Weather Research and Forecasting Model). We use these data to calculate the potential range of snow loading on the collapsed block, which is equivalent to a slope-parallel force of 7000-40,000 kN (Appendix A3). Considering the total precipitation between crack initiation (March 2016) and collapse (February 2021) this rises to $6.3 \times 10^9$ N to $9.9 \times 10^9$ N, or 2-3% of the total driving force of the collapse block.

In the absence of in-situ instrumentation and observations, it may not be possible to determine the exact cause of the failure at Chamoli. Nevertheless, we propose a mechanism which is compatible with both the lag between initial instability and collapse, and the timing of the eventual collapse. Snow and ice loading in the headwall crack would progressively increase the driving force of the collapse block, while meltwater infiltration and permafrost degradation in a bedrock fracture would steadily reduce its resistive forces (basal friction). The combination of these two processes would reduce the factor of safety and pre-condition the block for failure, with the early February positive temperature anomaly and loading from snowfall providing a final driver for mid-winter collapse.

## 4.3 Future perspectives : remote-sensing based hazard monitoring

Our work on the Chamoli avalanche took place after the collapse, with the full knowledge of the position of the avalanche source. This work is useful for better understanding the conditions of the slope collapse. However, to be directly useful for hazard monitoring and prevention, these techniques must identify avalanche locations and sizes before – rather than after – they occur. The key questions therefore remain: would it have been possible to identify the Chamoli landslide prior to its collapse using the methods used in our study, and can these methods be applied elsewhere to identify future failures?

Several factors suggest that the available pre-collapse data may have been useful for identifying the Chamoli rock-ice instability. Careful qualitative analyses of optical satellite images, feature tracking, and DEM analysis show clear precursory signs of slope failure around the Chamoli collapsed block. Satellite images show a crack growing over the 5 years prior to failure (Fig 3), feature tracking reveals tens of metres of horizontal displacement of the collapsed block, and DEM differences show tens of metres of vertical elevation change over the collapsed block. Combining this information with background knowledge about this region, such as the extreme relief, steep slopes, and historic avalanches, it would in principle have been possible to identify this as an unstable slope with high collapse potential.

While the data are sufficient to identify precursory signs of this rock-ice avalanche, there are important limitations to their use. The first key limitation is the very low signal to noise ratio of these data in the steep terrain most susceptible to slope failure. For feature tracking, the noise level of the composite 2016-2021 mean velocity maps is low (<1 m per year). However, the background noise level (as evaluated over stable bedrock) of individual velocity maps is much higher – and in some cases comparable to the magnitude of the signal (∼5-20 m per year). For the DEMs, artifacts range from metres to tens of metres in scale, and additional "noise" is introduced by real elevation changes from glacier and snowpack change (Fig 8b). While these issues with false positives can be mitigated, this is challenging without knowing the signal of interest.





InSAR, while also being susceptible to false positives, is additionally prone to false negatives. The north-facing aspect of Ronti peak provides a twofold challenge: the illumination of the slope is limited (low backscatter, Fig 6a), and any motion – assuming it is largely in the direction of the steepest slope – is oriented in the direction in which the radar instrument is least sensitive. Additionally, the non-glacierized area of the collapse wedge is small, making it challenging to identify fringe patterns amongst the noise. Furthermore, with the largest velocities reaching tens of meters per year, the InSAR measurements are prone to phase aliasing and underestimation of the true displacement. Sentinel-1 InSAR would not have provided an adequate tool for monitoring in this case, even with knowledge of the location of the instability.

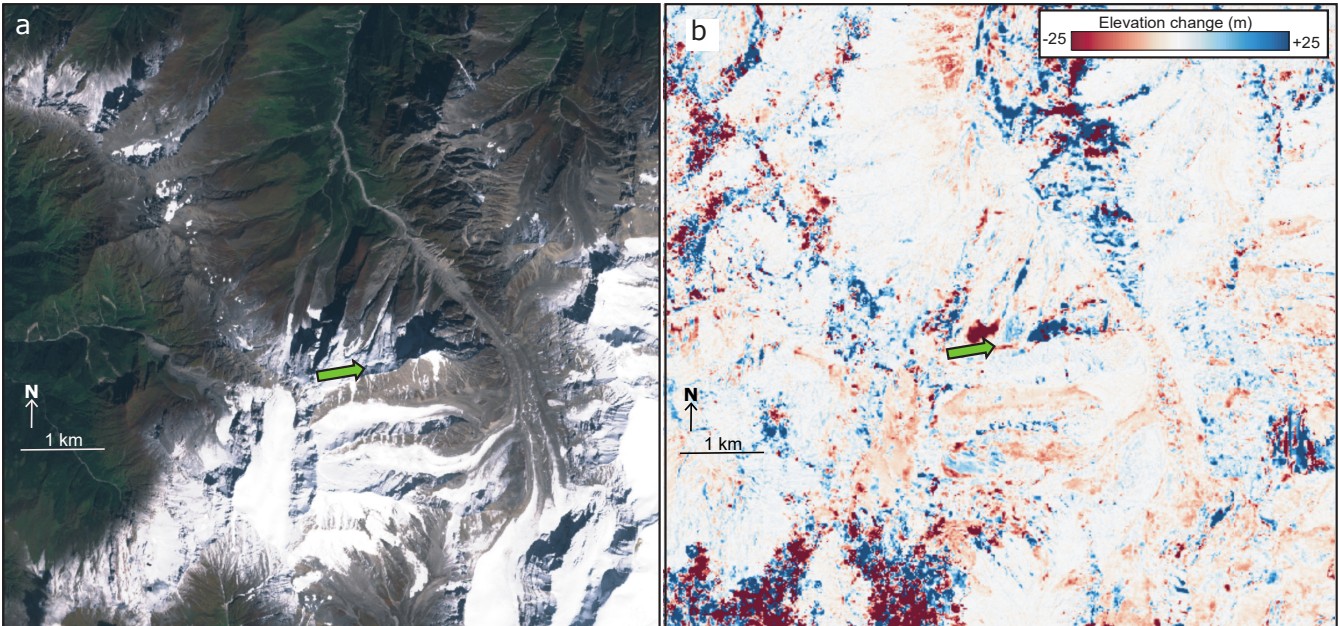

**Figure 8.** Optical satellite image (a: Sentinel-2; 28th of September 2020) and DoD (b; 2018-2015) of the Chamoli collapse site. Note the large number of steep slopes, complex terrain, and high noise levels in the DoD. In order to be useful for hazard prevention, these methods need to be able to identify potentially hazardous slopes without prior knowledge about collapses (e.g. green arrow shows the location of the headwall crack).

The second key limitation is that none of the datasets produced in this work could predict the timing of collapse. While most methods pick up precursory signs of slope failure, these begin almost five years prior to eventual collapse. In addition, the largest magnitude changes did not occur immediately prior to failure, but rather preceded failure by around three years. Even with the knowledge that the collapse occurred on 7 February 2021, there are no obviously anomalous signs that a failure was imminent in late 2020 or early 2021.

One final limitation is related to the immense size of hazardous areas relative to the scale of hazards themselves. The Chamoli collapsed block had an area of around 0.25 km$^2$, while the Himalaya cover over half a million km$^2$. Any methods aimed at automatically detecting hazards prior to their occurrence must have a low 'false positive' (identified as a hazard in the database,





but not of real concern) rate, or any resulting database will be populated primarily with these incorrectly flagged regions. This becomes a major challenge when considering the high incidence of noise and artifacts in feature-tracking derived displacement

or DoD maps (e.g. Figure 8).

Overall, forecasting the 7 February 2021 Chamoli rock-ice avalanche prior to its occurrence from remotely sensed datasets would have been very challenging, and certainly not routine work using well-established methods. Current image resolution, characteristics, and processing algorithms result in noise levels on a similar order to the signal itself – although joint interpretation of feature tracking results, DEM differences, and satellite images does reveal clear precursory signs of slope instability.

In addition, none of the data in this study are able to adequately forecast the timing of collapse. As such, current archives of satellite images do not currently appear to be practical for forecasting individual events. At the same time, this should not prevent remote monitoring of hazardous zones, particularly when adjacent to vulnerable areas. Every slope failure will exhibit a different range of pre-collapse signals, and new instabilities might be recognized in some cases. Even though the forecasting of individual events remains a challenge, these data have value for identifying zones of highest risk for in-situ monitoring or

the installation of early-warning systems (Cook et al., 2021).

Feature-tracking, DEM difference, and InSAR datasets can be processed and analyzed on a regional or even global scale – and in many cases pre-processed datasets are already available online (e.g. Morin et al., 2016; Gardner et al., 2018). While these pre-processed datasets are not generally produced for slope stability monitoring, they can be used to improve hazard maps and reduce landslide related damage. Future advances in Earth observation satellite capabilities and processing algorithms will

improve the quality of such products.

## 5    Conclusions

The deadly 7 February 2021 Chamoli rock-ice avalanche was initiated by failure of >25 Mm$^3$ of rock and ice high in the Uttarakhand Himalaya. We investigated the conditions of the avalanche source zone over the decades preceding collapse through a combination of optical and radar satellite images. We used feature tracking to calculate horizontal slope displacements, and

differenced photogrammetrically generated DEMs to investigate vertical displacements. We showed that the collapsed block moved 20-30 m prior to its collapse, with most rapid motion occurring around 3 years prior to failure. Comparison between our datasets and synthetic displacement maps shows that the motion occurred primarily via down-slope translation, opening up a deep crack at the headwall. A combination of permafrost degradation and snow and ice debris loading within this headwall crack may explain both the lag between initial instability and collapse, and the mid-winter timing of the collapse. Finally,

we assessed the potential of these datasets and approaches for monitoring other unstable slopes. While they were effective at identifying precursory signals at a known collapse site, it remains very challenging to predict such collapses with sufficient levels of confidence in high-mountain areas.





*Code and data availability.* All code used in this study is openly available online. GIV can be downloaded from https://github.com/MaxVWDV/glacier-image-velocimetry, AutoRIFT from https://github.com/nasa-jpl/autoRIFT, ASP from https://github.com/NeoGeographyToolkit/StereoPipeline,

and ISCE from https://github.com/isce-framework/isce2. The 2015 pre-event DEM is available at https://doi.org/10.5281/zenodo.4554646, and the 2021 post-event DEM at https://doi.org/10.5281/zenodo.4558691.

## Appendix A: Pre-event DEM mosaics

### A1 CartoSat-1 (2017/2018)

We procured four CartoSat-1 stereo pairs from October 2017 and November 2018 (Supplementary Data sheet) to compute

DEMs for an intermediate period between 2015 to 2021. Initial assessments of the CartoSat-1 products revealed high stereo ray intersection errors ($> 100$ m) and offsets from reference elevation models ($\sim 400$ m), indicative of poor relative and absolute accuracy of the vendor supplied RPC models. To address these issues we employed ASP's bundle_adjust utility on all the eight overlapping images and the corresponding RPC models using similar techniques as described in Bhushan et al. (2021); Dehecq et al. (2020). The bundle adjustment procedure matches similar features between all input overlapping

images and minimises their reprojection error by updating the RPC camera with translation and rotation parameters. Using the updated RPC model obtained after bundle adjustment, we generated a draft DEM from one of the four pairs using the default ASP settings and aligned it to a filtered and masked version of the HMA 8 m DEM mosaic v2 (Shean, 2021). The alignment matrix was used to further update the self-consistent RPC model output from bundle adjustment, ensuring improved absolute geolocation accuracy. Following this, the input images were orthorectified at their native resolution of 2.5 m using the 30 m

Copernicus DEM (converted to ellipsoidal heights) and stereo processing (correlation and triangulation) was performed for all the four input pairs using the settings described in Shean and Bhushan (2021).

The CartoSat-1 DEMs were posted at 10 m resolution with UTM 44N projection and heights above the WGS84 ellipsoid. Consequently, the DEMs were co-registered to the HMA 8 m DEM mosaic v2 (Shean, 2021) over non-glacierized surfaces using a two step procedure: ASP's pc_align followed by Nuth and Kääb (2011) alignment implemented in Shean et al. (2019)]

to remove any residual horizontal and vertical offsets in the final output DEMs.

### A2 SPOT-7 (2018)

We also derived a DEM from the December 24, 2018 SPOT-7 stereo pair using ASP's Semi Global Matching correlator and other settings similar to those described in Lacroix (2016); Deschamps-Berger et al. (2020). The final output DEM was posted at a resolution of 10 m with UTM 44N projection and heights above the WGS84 ellipsoid. The DEM was co-registered to

the HMA 8 m DEM mosaic v2 (Shean, 2021) over non-glacierized surfaces to ensure consistency with all the DEM products derived in this study.





**Appendix B: Factor of safety calculations for the Chamoli bloc**

The factor of safety $FS$ is calculated from the balance driving and resistive forces (e.g. Záruba and Mencl, 2014; Das and Sivakugan, 2016).:

$$FS = \frac{F_R}{F_D} = \frac{AC + Mg\cos(\alpha)\tan(\phi)}{Mg\sin(\alpha)}$$
(B1)

In which $A$ is slip surface area, $C$ is cohesion, $M$ is the mass of the unstable region, $g$ is gravity, $\alpha$ is slope, and $\phi$ is the friction angle. A system may be considered unstable when the factor of safety falls below 1.

Introducing an additional horizontal force $F_H$ modifies this balance in two ways: firstly by increasing the driving force, and secondly by altering the angle between the driving force vector and resistive forces vector:

$$FS = \frac{AC + Mg\cos(\alpha + \alpha')\tan(\phi)}{F_H + Mg\sin(\alpha + \alpha')}$$
(B2)

The change in angle of the driving force vector $\alpha'$ is then given by $\alpha' = \arctan(\frac{F_H}{Mg\sin(\alpha)})$. In our situation, for a given mass accumulated in the headwall crack $M_C$ we have $F_H = M_C\sin(\alpha)$.

The pre-event storm brought 8.5 to 48 mm water equivalent of precipitation (Shugar et al. (2021); estimates from local weather stations and Weather Research and Forecasting Model). We may use this data to calculate possible loading of this
snow on the collapsed block - considering a 500 m long, 70 m wide crack with a 500 m long and fed by a 180 m wide avalanche zone. Assuming that all of the snowfall was channeled into the crack, total loading $M_C$ would be equal to:

$$M_C = A_A * P * \rho_P * g$$
(B3)

With $A_A$ being the accumulation area feeding the crack, $P$ being precipitation (in metres), $\rho_P$ being the density of the precipitation. Total snow loading in the headwall crack associated with this single precipitation event would therefore be
10000-60000 kN, equivalent to a slope-parallel horizontal force of 7000-40000 kN.

GPM IMGERG precipitation data suggests that around $9 \pm 2$ m of precipitation fell in the collapse area between crack initiation in 2016 and collapse in 2021. Using the same calculation, maximum snow load in the headwall crack is equal to 8.6-13.5 $\times 10^9$ N, equivalent to a slope-parallel horizontal force of 6.3-9.9 $\times 10^9$ N. For reference, the estimated total driving force of the collapse bloc, composed of 21 Mm$^3$ of rock and 6 Mm$^3$ of ice, is  $4.0 \times 10^{1}1$ N.

*Author contributions.* All authors designed the study and conducted the research. MVWDV wrote the paper, with input from all co-authors. The final version has been approved by all co-authors.

*Competing interests.* The authors declare no competing interests.



*Acknowledgements.* MVWDV was funded by a University of Minnesota College of Science and Engineering fellowship and a Doctoral Dissertation Fellowship. DHS was funded by Natural Sciences and Engineering Research Council of Canada (NSERC) Discovery Grant
2020-04207. SG and EB acknowledge funding from the French Space Agency (CNES). SG received funding from the Programme National de Télédétection Spatiale (PNTS grant no. PNTS-2018-4). AK acknowledges support from the ESA Glacier CCI project (grant no. 4000109873/14/I-NB). MJ was funded by the WSL research program Climate Change Impacts on Alpine Mass Movements (CCAMM) and the Swiss National Science Foundation (grant no. 200021_184634). SB was supported by a NASA FINESST award (80NSSC19K1338). DES was supported by a NASA HiMAT-2 award (80NSSC20K1595).





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
