# Peer review of "Pre-collapse motion of the February 2021 Chamoli rock-ice avalanche, Indian Himalaya"

_Natural Hazards and Earth System Sciences, 2021_

## Author Comment (AC1)

**Pre-collapse motion of the February 2021 Chamoli rock-ice avalanche, Indian Himalaya**

Maximillian Van Wyk de Vries[1,2], Shashank Bhushan[3], Mylène Jacquemart[4,5], César Deschamps-Berger[6], Etienne Berthier[7], Simon Gascoin[6], David E. Shean[3], Dan H. Shugar[8], and Andreas Kääb[9]

[1]St Anthonys Falls laboratory, University of Minnesota, Minneapolis, MN, USA
[2]Department of Earth and Environmental Sciences, University of Minnesota, Minneapolis, MN, USA
[3]Department of Civil and Environmental Engineering, University of Washington, Seattle, WA, USA
[4]Laboratory for Hydraulics, Hydrology, and Glaciology (VAW), ETH Zurich, Zurich, Switzerland
[5]Swiss Federal Institute for Forest, Snow, and Landscape Research (WSL), Birmensdorf, Switzerland
[6]CESBIO, Université de Toulouse, CNRS, CNES, IRD, INRAE, UPS, Toulouse, France
[7]LEGOS, Université de Toulouse, CNES, CNRS, IRD, UPS, Toulouse, France
[8]Water, Sediment, Hazards, and Earth-surface Dynamics (waterSHED) Lab, Department of Geoscience, University of Calgary, Canada
[9]Department of Geosciences, University of Oslo, Oslo, Norway

Correspondence: Maximillian Van Wyk de Vries (vanwy048@umn.edu)

**Response to Reviews**

**Review 1**

Dear Authors,

The manuscript is written well and address the research questions defined by you. However, I have some suggestions/comments before the manuscript is accepted for publication in NHESS.

We thank the reviewer for these comments, and for their positive assessment of our manuscript. We have responded to the comments in detail below, with the review comments given in black and our responses in red.

Himalayan terrain many times induces decorrelations. due to its topography as well as vegetation. The authors have tried to implement simple DInSAR methodology and were unable to obtain good interferograms. Loss of coherence is the main challenge to the InSAR application in the area.
I am afraid that simple PSI technique will hardly yield any significant result. Even the techniques such as SBAS, SqueeSAR etc may also fail to produce anything. As a suggestion you can try A-DinSAR techniques such as SBAS or techniques based on distributed scatterers such as Quasi PS (Perissin & Wang, 2011; Razi et al., 2018) or SqueeSAR (Ferretti et al., 2011) in this region.

Perissin, D., & Wang, T. (2011). Repeat-pass SAR interferometry with partially coherent targets. IEEE Transactions on Geoscience and Remote Sensing, 50(1), 271-280.

Razi, P., Sumantyo, J. T. S., Perissin, D., Febriany, F., & Izumi, Y. (2018, August). Multi-temporal land deformation monitoring in V shape area using quasi-persistent scatterer (Q-PS) interferometry technique. In 2018 Progress in Electromagnetics Research Symposium (PIERS-Toyama) (pp. 910-915). IEEE.

Ferretti, A., Fumagalli, A., Novali, F., Prati, C., Rocca, F., & Rucci, A. (2011). A new algorithm for processing interferometric data-stacks: SqueeSAR. IEEE transactions on geoscience and remote sensing, 49(9), 3460-3470.

A study has been done like this on the same study area. They have claimed that they have used PSI on the regional level.
Kothyari, G. C., Joshi, N., Taloor, A. K., Malik, K., Dumka, R., Sati, S. P., & Sundriyal, Y. P. (2021). Reconstruction of active surface deformation in the Rishi Ganga basin, Central Himalaya using PSInSAR: a feedback towards understanding the 7th February 2021 Flash Flood. Advances in Space Research.

Thank you for pointing out the difficulties of performing InSAR measurements in such challenging terrain. As we note in our manuscript, shadowing and decorrelation are particularly problematic on this face. The primary issue is that the release zone is in radar shadow in the images of the ascending orbit and is affected by layover in the descending orbit. The additional tools that you mention (PSInSAR, SBAS, SqueeSAR etc.) are all time-series analysis tools. We did not attempt to generate a time-series for the Chamoli failure because the data quality is too low. In other words, too few coherent pixels are present in the failure area to allow for the extraction of a time-series or the generation of a stack, therefore we cannot apply time-series tools in this case.

Kothyari et al. (2021) do indeed present an attempt to reconstruct regional displacements using PSI. However, their regional map of cumulative ground displacement (Figure 8b) does not show any exceptional signal at the site of the Chamoli rock-ice avalanche. Instead, their results exhibit ~40 mm anomalies across the entire study area, suggesting that noise dominates over any signal. Their cumulative displacement timeseries also show no long-term pattern consistent with an unstable block, instead fluctuating around zero (as expected for noise). This is compatible with our finding that the data quality is too low to detect the displacement signal

For Figure 6b, I suggest the authors check all the available interferograms to confirm that the two areas marked with "Rock glacier motion" are always moving. From only one interferogram, it is hard to say the motion. The area marked with "Atmospheric noise" seems not so evident because there are some strange values in the surrounding area possibly caused by unwrapping errors due

to the low coherence. If these areas can't be confirmed from other interferograms, I suggest writing these areas are possible rock glacier motion or possible atmospheric noise. I suggest also writing "2.8 cm" in the caption of Figure 6 after "wrapped phase".

Thank you for this comment. The rock glacier motion was indeed labeled as such after the analysis of all available interferograms, as well as the study of optical images that identified the landscape features as such. The area denoted as "Atmospheric noise" cannot be caused by unwrapping errors, since our figure shows wrapped phase which has not yet been unwrapped. However, the cause of this specific fringe pattern is somewhat unclear, and we agree that labeling it with "possible atmospheric noise" is preferable. We have also implemented your suggestion for the addition to the caption.

In figure 6b, I'm not completely sure about the "atmospheric noise". I suggest to check the displacement time series, if available, because atmospheric noise can be identified as strange peaks.

We agree with this comment and have responded to it in more detail above.

---

## Author Comment (AC2)

**Pre-collapse motion of the February 2021 Chamoli rock-ice avalanche, Indian Himalaya**

Maximillian Van Wyk de Vries[1,2], Shashank Bhushan[3], Mylène Jacquemart[4,5], César Deschamps-Berger[6], Etienne Berthier[7], Simon Gascoin[6], David E. Shean[3], Dan H. Shugar[8], and Andreas Kääb[9]

[1]St Anthonys Falls laboratory, University of Minnesota, Minneapolis, MN, USA
[2]Department of Earth and Environmental Sciences, University of Minnesota, Minneapolis, MN, USA
[3]Department of Civil and Environmental Engineering, University of Washington, Seattle, WA, USA
[4]Laboratory for Hydraulics, Hydrology, and Glaciology (VAW), ETH Zurich, Zurich, Switzerland
[5]Swiss Federal Institute for Forest, Snow, and Landscape Research (WSL), Birmensdorf, Switzerland
[6]CESBIO, Université de Toulouse, CNRS, CNES, IRD, INRAE, UPS, Toulouse, France
[7]LEGOS, Université de Toulouse, CNES, CNRS, IRD, UPS, Toulouse, France
[8]Water, Sediment, Hazards, and Earth-surface Dynamics (waterSHED) Lab, Department of Geoscience, University of Calgary, Canada
[9]Department of Geosciences, University of Oslo, Oslo, Norway

**Correspondence:** Maximillian Van Wyk de Vries (vanwy048@umn.edu)

**Response to Reviews**

**Review 2**

We thank the reviewer for their comments and suggestions on our manuscript. We have responded to the comments in detail below, with the review comments given in black and our responses in red. While we disagree with the reviewer's assertion that our manuscript is outside of NHESS's scope, we are grateful for the reviewer's questions and implement changes to our manuscript in line with these.

The manuscript focuses on investigating pre-failure surface displacement for the case study of the 2021 Chamoly avalanche. The case is undoubtedly interesting but, in my opinion, the manuscript should not be considered for publication in NHESS. In the aims and scope of the journal it is stated that "The following are generally considered out-of-scope or we do not encourage: [...] Localised case studies with no broader implications (in other words, ask yourself, what would someone else in another region learn from the case study that you have done; what is the broader context?)." While I recognise that the authors' work could have broader implications, these are not discussed at all in the manuscript. The manuscript is indeed completely focused on the case study and the authors make indeed conclusions (e.g., the unpredictability of the timing of collapse) only for the case study and do not discuss what they learned in terms of general implications.

We thank the reviewer for acknowledging the potential interest in our study but disagree that this manuscript is outside the scope of NHESS.

Reviewing the scope of the journal, we see the following three points:

- the study of the evolution of natural systems towards extreme conditions, and the detection and monitoring of precursors of the evolution;

- the detection, monitoring, and modelling of natural phenomena, and the integration of measurements and models for the understanding and forecasting of the behaviour and the spatial and temporal evolution of hazardous natural events as well as their consequences;

- the design, development, experimentation, and validation of new techniques, methods, and tools for the detection, mapping, monitoring, and modelling of natural hazards and their human, environmental, and societal consequences;

We believe that our manuscript, through an in-depth study of the Chamoli rock-ice avalanche pre-collapse conditions and the methods used to assess this instability, contributes to all three of these points. Furthermore, we do not believe that the reviewer's statement "While the authors' work could have broader implications, these are not discussed at all" is accurate. Even in its present state, our objectives are broad enough to be of relevance beyond the Chamoli rock-ice avalanche (e.g. our objective line 110: "Would these pre-collapse datasets and tools would be adequate to identify this hazardous slope without the prior knowledge of its failure?"), and we dedicate an entire subsection of our Discussion to the broader implications ("Future perspectives : remote-sensing based hazard monitoring").

Indeed, one of our primary aims in this manuscript is to use the complex, but also well-studied and information-rich Chamoli rock-ice avalanche to evaluate the potential of Earth Observation datasets to detect unstable slopes and landslides prior to their occurrence. When discussed from this perspective, we believe that detailed individual case studies can contribute to our understanding of hazard monitoring techniques and processes well beyond the study area. We discuss below how we have clarified and, in some cases, expanded on this aspect of our manuscript.

In summary, our detailed analysis of the Chamoli rock-ice avalanche draws conclusions that are both novel and more widely relevant for the investigation of the pre-collapse conditions of other unstable slopes, particularly in complex high-relief or glaciated environments.

A review of recently published papers in NHESS shows many examples using a detailed local case study to draw wider conclusions, for instance:
Submarine landslide source modeling using the 3D slope stability analysis method for the 2018 Palu, Sulawesi, tsunami,
The Cambodian Mekong floodplain under future development plans and climate change,

[Spatiotemporal evolution and meteorological triggering conditions of hydrological drought in the Hun River basin, NE China](), [Geo-historical database of flood impacts in Alpine catchments (HIFAVa database, Arve River, France, 1850–2015)](), or [Correlation of wind waves and sea level variations on the coast of the seasonally ice-covered Gulf of Finland]() (all published within the past month at the time of writing).

Finally, we also note that our manuscript passed the editorial access review editor which suggests that it is potentially of interest for the NHESS readership "Manuscripts submitted to NHESS at first undergo a rapid access review by the editor [...] to identify and sort out manuscripts with obvious deficiencies in view of the above principal evaluation criteria. [...] the paper should contribute something new and interesting to the community." (https://www.natural-hazards-and-earth-system-sciences.net/peer_review/review_criteria.html)

As mentioned above, we do believe that valuable points were made in this review. We therefore propose to build on these questions listed below to finalize our manuscript:

Could the timing have been predicted if more images were available (e.g., 24 prior to the collapse, 8 hours prior to the collapse, etc.)?

We agree that this is an important point and will further develop it in the discussion of our revised manuscript through a discussion of what is visible in the pre-collapse imagery, and what would need to be visible to yield useful insight into the timing of collapse.

To specifically answer this question, we do not believe that the data are available to provide a meaningful answer. This question could perhaps have been answered if any image had been collected in the hours preceding collapse, but no such image exists. We do not detect any anomalous pre-collapse increase in velocity in our Sentinel-2 feature tracking timeseries, with the final pre-collapse image pair being the 31st of January and 5th of February 2021 (8 days and 2 days prior to collapse), although the precision of displacements derived from any single image pair is low.

Furthermore, even if the collapse timing were detectable in pre-collapse imagery, this would not in itself be sufficient for the methods to be applied for hazard monitoring. For methods to be useful in hazard monitoring, the timing of collapse must be identifiable without special attention to the imagery immediately preceding collapse, as this timing will not be known.

Is it just a matter of noise or, even without noise, no trend to failure could be seen? Is it a matter of resolution, instead?

Noise and resolution are not independent in the case of feature tracking or DEM generation and differencing. For instance, in feature tracking, the choice of a larger window size will generally reduce noise levels at the cost of lower resolution. A related question could therefore be 'For a spatial resolution sufficient to study a given unstable slope, is the signal-to-noise ratio low enough for the hazard to be monitored?' or conversely 'Is the minimum spatial resolution achievable while maintaining a certain signal- to-noise ratio sufficient for monitoring a given hazard?'. The use of different satellites with higher resolution can provide an improvement, but does not always do so: for instance, the 3 m resolution Planet imagery often produces velocity maps with higher noise levels than the 10m resolution Sentinel-2 imagery due to greater differences in imaging geometry between image pairs.

We discuss some of the limitations of current noise levels in the final section of our discussions and in Figure 8.

Is it a matter of mechanism of failure (e.g., a very steep tertiary creep that causes orders of magnitude of acceleration in a matter of minutes/hours?). Is this mechanism rare or typical in such a type of failures?

The exact collapse mechanism is secondary to our main research question of whether this collapse could have been detected or forecast from remotely sensed data. There is no geomechanical data available in the case of the Chamoli collapse, and any assessments of the mechanisms of failure must be conducted using remote datasets (including seismic data; https://www.nature.com/articles/s41598-022-07491-y) and post-event field investigations, complemented by numerical modeling or analogy to other, better instrumented, areas. Here, we propose a collapse mechanism compatible with the available data, based on a combination of permafrost thaw and loading in the headwall crack.

What is the general conclusion in terms of remote sensing capability in predicting failures/timing of failures? What types of landslides could be predicted, instead? Are there examples in the literature of successful/unsuccessful predictions of other case studies based on similar data sources? Will we ever be able to predict the timing of failure based on satellite remote sensing alone?

The final two sentences of our conclusions provide a general conclusion in terms of remote sensing capability in predicting failures/timing of failures. For convenience, we include them here:
"Finally, we assessed the potential of these datasets and approaches for monitoring other unstable slopes. While they were effective at identifying precursory signals at a known collapse site, it remains very challenging to predict such collapses with sufficient levels of confidence in high-mountain areas. "

We agree that the above questions are important, but also note that many are outside of the scope of our paper. We cannot and should not conclude in general terms about whether remote sensing is able to predict the timing of failure, as each individual site is subject to unique conditions. We found that identifying the timing of collapse at Chamoli was likely not possible. It may, however, be possible in other locations using comparable datasets.

We believe that the following paragraphs in our discussions section (lines 376-385) provides some answers to these questions, and could be further developed in a revised version of this manuscript:

"Overall, forecasting the 7 February 2021 Chamoli rock-ice avalanche prior to its occurrence from remotely sensed datasets would have been very challenging, and certainly not routine work using well-established methods. Current image resolution, characteristics, and processing algorithms result in noise levels on a similar order to the signal itself – although joint interpretation of feature tracking results, DEM differences, and satellite images does reveal clear precursory signs of slope instability. In addition, none of the data in this study are able to adequately forecast the timing of collapse. As such, current archives of satellite images do not currently appear to be practical for forecasting individual events. At the same time, this should not prevent remote monitoring of hazardous zones, particularly when adjacent to vulnerable areas. Every slope failure will exhibit a different range of pre-collapse signals, and new instabilities might be recognized in some cases. Even though the forecasting of individual events remains a challenge, these data have value for identifying zones of highest risk for in-situ monitoring or the installation of early-warning systems (Cook et al., 2021)."

Feature-tracking, DEM differencing, and InSAR datasets can be processed and analyzed on a regional or even global scale – and in many cases pre-processed datasets are already available online (Morin et al., 2016, Gardner et al., 2018). While these pre-processed datasets are not generally produced for slope stability monitoring, they can be used to improve hazard maps and reduce landslide related damage. Future advances in Earth observation satellite capabilities and processing algorithms will improve the quality of such products."

In addition, with reference to the specific case study, I found the discussion speculative, for instance, when it came to the safety factor as all discussions on driving and resisting forces were based on general knowledge/speculation and not supported by, e.g., geomechanical data of the case study.

As we note above, no geomechanical data or other ground-based assessments of the conditions of the avalanche block are available. We are aware of this limitation and note in our Discussion section that: "In the absence of in-situ instrumentation and observations, it may not be possible to

determine the exact cause of the failure at Chamoli.". Nevertheless, our proposed collapse mechanism is compatible with our remotely sensed data and is able to explain the mid-winter timing of collapse.

Also, the introduction is too generic, describing information that is very well known to researchers in the field. Perhaps the introduction could have focused only on recent advances in remote sensing techniques for natural hazards that are perhaps closing a knowledge gap and enabling the type of analysis conducted by the authors, albeit with remaining limitations.

We appreciate this comment about our introduction. We had aimed to provide a broad overview of each technique we had used such that our paper was more accessible to researchers beyond this specific field of study. We acknowledge that more specific information would be useful, particularly about how these techniques are currently used for landslide monitoring and will revise this section accordingly.

---

## Author Response (AR2)

**Pre-collapse motion of the February 2021 Chamoli rock-ice avalanche, Indian Himalaya**

Maximillian Van Wyk de Vries[1,2,3], Shashank Bhushan[4], Mylène Jacquemart[5,6], César Deschamps-Berger[7], Etienne Berthier[8], Simon Gascoin[7], David E. Shean[4], Dan H. Shugar[9], and Andreas Kääb[10]

[1]St Anthonys Falls laboratory, University of Minnesota, Minneapolis, MN, USA
[2]Department of Earth and Environmental Sciences, University of Minnesota, Minneapolis, MN, USA
[3]School of Environmental Sciences, University of Liverpool, Liverpool, L3 5DA, UK.
[4]Department of Civil and Environmental Engineering, University of Washington, Seattle, WA, USA
[5]Laboratory for Hydraulics, Hydrology, and Glaciology (VAW), ETH Zurich, Zurich, Switzerland
[6]Swiss Federal Institute for Forest, Snow, and Landscape Research (WSL), Birmensdorf, Switzerland
[7]CESBIO, Université de Toulouse, CNRS, CNES, IRD, INRAE, UPS, Toulouse, France
[8]LEGOS, Université de Toulouse, CNES, CNRS, IRD, UPS, Toulouse, France
[9]Water, Sediment, Hazards, and Earth-surface Dynamics (waterSHED) Lab, Department of Geoscience, University of Calgary, Canada
[10]Department of Geosciences, University of Oslo, Oslo, Norway

Correspondence: Maximillian Van Wyk de Vries (vanwy048@umn.edu)

**Response to minor revision**

*Editor comment*

dear authors

after receiving the review reports on your manuscript, I am glad to inform you that your paper can be accepted by the journal should you be willing to apply some minor revision and modifications to it.
In the attached, you will find the two reports.
Please try to answer to all reviewers' comments and suggestions on a point-by-point basis, as well as please modify the original document accordingly and attach all required docs to the submission.
After your modifications, the manuscript will be subject to a final revision by the Editor, without passing a further review stage.

Looking forward to receiving your amended manuscript and replies,
best wishes
FC

We thank the editor and both reviewers for taking the time to read through our manuscript, and are glad that the revised version of our manuscript is suitable for publication in NHESS. We respond in detail to comments from reviewer 2 below (reviewer 1 recommended acceptance

without providing any new comments), including a detailed note about the accessibility of very high resolution imagery. We write our comments in red, with reviewer comments in black.

***Review of Van Wyk de Vries et al., 'Pre-collapse motion of the February 2021 Chamoli rock-ice avalanche, Indian Himalaya'.***

Van Wyk de Vries et al. present an analyses of a variety of remotely sensed datasets to study the dynamics of the slope instability which produced the Chamoli ice-rock avalanche in Feb. 2021. Their results show how the broader Chamoli site has irregularly produced additional (smaller) ice and/or rock avalanches over the last few decades, and that the slope itself became mobile several years before the collapse event. Their results emphasize the strength of remotely sensed data to study processes operating over extended temporal baselines, and the authors nicely illustrate and discuss the limitations of their data where signal to noise ratios are less favorable. The authors present a synthesis of observations from other work on the Chamoli event and propose a trigger mechanism for the avalanche.

In my opinion the work is of high quality and the authors have carefully considered their evaluation of the application (and limitations) of remotely sensed data to study high mountain hazards such as ice/rock avalanches. I have a few minor comments about how the paper could be improved, which are listed below. Several of these comments relate to the over-emphasis of the capability of truly open access remotely sensed datasets in the study of these processes, which I think the authors would be wise to clarify, as this study certainly does not rely solely on publicly available datasets. Overall (noting previous reviewers comments) I find the work well within the scope of NHESS.

We thank the reviewer for their comments and for their positive overall view of our manuscript. We hope that the responses below and minor edits to our manuscript cover any remaining comments.

We respond to the comment about open access remotely sensed data below, but do not agree that its potential has been overstated. While this study also incorporates non open source data and demonstrates the value of merging multiple data sources, many of the core findings could have been reached with open source data alone (specifically, feature tracking of Sentinel-2 data clearly shows the pre-collapse displacement).

We do hope that comercial data becomes more readily available for the mitigation of natural hazards, and that new open-source monitoring missions provide comparable datasets in the future. Presently available open source data does, nevertheless, provide many new opportunities for hazard monitoring. We have added a comment about this in the manuscript lines 112-114

"However, most very high resolution imagery is not open source and is expensive to procure, limiting its use. Increased availability of this commercial imagery and/or new open-source stereo imagery satellites would provide many new opportunities for hazard monitoring."

And further edits in the discussions section lines 333-334

"In addition, the DEMs and elevation change maps used in this study were generated from imagery not accessible in open source archives. Changes in the accessibility of commercial data or the launch of new, open access, stereo-imagery satellites would facilitate the use of elevation change in large-scale geohazard monitoring."

Minor comments:

Title: 'Indian Himalaya' could be narrowed down to the 'Garhwal Himalaya' I think.

After some thought, we have retained 'Indian Himalaya', as the name 'Chamoli' already provides a narrower geographic area and additional information about the more general area is useful to those not familiar with the region.

L43- I think the authors need to be careful not to oversell the capabilities of truly 'open access' data here (and later in the manuscript). Many of the datasets the authors have used are not 'open access' as things currently stand (the high and very high resolution stereo imagery have not been sourced from open archives) and the analyses would not have taken the same shape without them. It would be inaccurate to suggest that the same kind of data could be easily acquired to study different events. It may even be a point for discussion for later in the manuscript to emphasize how the study of these events relies on high resolution, high precision data which are not abundant in the public domain.

We agree with the reviewer that not all data used in this paper is open access, but are not implying this in the sentence referred to here ("Growing archives of high-resolution, open access Earth observation data remain largely untapped for landslide monitoring."). The success of Sentinel-2 derived displacements in this study is one example of this, and other studies have successfully leveraged Sentinel-1 derived InSAR maps for displacement mapping.

As Figure 7b shows, even change maps derived from very high resolution data may have limitations for large-scale analysis, and improvements in remote monitoring of slope stability need not only come from higher resolution imagery. We are not suggesting that the data used in our study could be used for any event in the world, but simply that a large volume of truly open source data is now available on a global scale.

L70- The 80-20 composition statement could do with a citation, otherwise it just reads as some kind of informed guess.

The calculations behind this compositional breakdown are provided in Shugar et al., 2021. We have added a citation on this sentence.

L85- As with my comment on L43, this is at odds with the assertion that high-resolution, open access imagery is available to study these events.

We agree with the reviewer that very high resolution imagery is not open access in most cases. We have added some additional discussion about this in other sections as shown in our responses to the other points.

The majority of our feature tracking results are derived from Sentinel-2 imagery, which is truly open access. The calculations made with the other datasets primarily serve as validation of this data (e.g. Figure 4 d-g). Given that our main feature tracking conclusions can be derived entirely from open-source data, we consider the text to be appropriate in this section.

L98- Might be worth tailoring this point about sensing in conditions inhibitive to optical imagery to emphasise the benefits of InSAR in the monsoon period, perhaps?

We have added a note about the monsoon to this sentence, and agree that this is particularly relevant in the context of rainfall-triggered landslide motion.

L108- Again, high-resolution imagery suitable for tasks such as this come at a high price. I think its worth mentioning that these very high resolution stereo imagery were specifically acquired for the AOI, not just plucked from an open archive.

We have added an extra note to this sentence about the high cost of this UHR imagery. Not all of the imagery we used was specifically tasked over this AOI (only the 2021 ones), and there are large public datasets derived from UHR imagery (e.g. ArcticDEM). However, these are not always available and we agree that the cost is useful to note.

We have also, as noted above, added a comment about this in the manuscript lines 112-114

"However, most very high resolution imagery is not open source and is expensive to procure, limiting its use. Increased availability of this commercial imagery and/or new open-source stereo imagery satellites would provide many new opportunities for hazard monitoring."

L154- delete 'is' after AutoRIFT.

We have removed 'is'.

L174- this --> these velocity data as they are plural?

We have changed 'this' to 'these' as recommended.

L175- Is there a Figure the authors can refer to in this paragraph?

We have added a reference to figure 1 which shows this bedrock ridge.

L217- Should this be 'as a real sign' rather than 'is a real sign'?

This should be 'as'. We have corrected the typo.

L227- A trivial point, but it can't have reached its max. width by the end of 2018 if it 'widened further' between 2018 and 7th feb, 2021, can it?

We have changed this to read 'The crack grew until the end of 2018'.

Figures 2-4- really nice!
Only thought…do you need the inset map of India again on Figure 4? Just seems to hinder your formatting slightly.

We thank the reviewer for their positive comment on our figures. We agree that the inset was not necessary due to Figure 1, and have removed it to improve the formatting of this figure.

Figure 6- A few minor points:

-Are the elevation change maps in panels B and C derived using the SPOT, Cartosat or composite 2018 DEM in combination with the pre-event and post-event DEMs?

The composite DEM was used. We have added this to the figure caption for clarity.

-Is there any way that the same elevation change range can be used in panels B-E? +/-25, +/-100 and +/-40 are all used here and it's not that easy to interpret.

We selected these different elevation change ranges as they are necessary for the details to be visible in each case. We could use +/- 100 m for all four panels, but the details of the precollapse change would be difficult to visualize. We have added a note to the caption that the elevation change bounds are different.

-The differences over stable ground areas seem to have opposite signs in panels B and C. Is this the result of some sort of minor bias in the 2018 DEM?

The systematic differences over stable ground are likely related to differences in snow cover between the three time periods. These differences are, however, small enough to not affect the signal of interest.

Figure 7- Is it worth adding some glacier outlines to this figure? The preceding text refers to elevation change signal over glaciers and it'd be easier to pick this out of the elevation change map on the right hand side with the help of outlines.

Our primary objective in this figure was to show the difficulty in separating the block displacement signal from the 'noise' (both glacier elevation change and DEM errors). We therefore do not add glacier outlines, as we consider that these may distract from this point.

L330- The modeling here is described very briefly and I doubt could be replicated with the details provided. Is this modeling based on an established method?

This 'model' is very simple and involves a simple translation of one elevation raster (collapsed block thickness) over another (post collapse topography). We have added more details to this paragraph (L335-340) to clarify this point.

L350- What period is referred to as 'longer term' here? Can the authors provide an estimate of the warming that has occurred over the last 2-3 decades in the region? This might provide a little more context for the discussion of freeze-thaw fracturing and permafrost degradation which takes place in the paragraphs below.

We have added more information to this sentence, including an estimated total warming over the past 3 decades. This sentence now reads:
"On the longer term, this region has warmed an estimated 0.014 K (Zhou et al., 2021) to 0.033 K (Shrestha et al., 2021) per year since 1980, for a total warming of 0.4 K to 0.9 K over the past three decades."

L360- Refer to Appendix B in here somewhere?

We have added a reference to this appendix here (L360-361).